# Behavioral Despair Is Blocked by the Flavonoid Chrysin (5,7-Dihydroxyflavone) in a Rat Model of Surgical Menopause

**DOI:** 10.3390/molecules28020587

**Published:** 2023-01-06

**Authors:** Luis Ángel Flores-Aguilar, Jonathan Cueto-Escobedo, Abraham Puga-Olguín, Oscar Jerónimo Olmos-Vázquez, Gilberto Uriel Rosas-Sánchez, Blandina Bernal-Morales, Juan Francisco Rodríguez-Landa

**Affiliations:** 1Programa de Doctorado en Neuroetología, Instituto de Neuroetología, Universidad Veracruzana, Xalapa 91190, Mexico; 2Departamento de Investigación Clínica y Traslacional, Instituto de Ciencias de la Salud, Universidad Veracruzana, Xalapa 91190, Mexico; 3Unidad de Salud Integrativa, Centro de EcoAlfabetización y Diálogo de Saberes, Universidad Veracruzana, Xalapa 91060, Mexico; 4Laboratorio de Neurofarmacología, Instituto de Neuroetología, Universidad Veracruzana, Xalapa 91190, Mexico

**Keywords:** anti-stress, benzodiazepine, despair, flavonoid, neurosteroid, surgical menopause

## Abstract

Women have a high susceptibility to the negative effects of stress. Hormonal changes experienced throughout their reproductive life partially contribute to a higher incidence of anxiety and depression symptoms, particularly, during natural or surgical menopause. In preclinical research, the flavonoid chrysin (5,7-dihydroxyflavone) exerts anxiolytic- and anti-despair-like effects; however, it is unknown whether chrysin exerts a protective effect against the behavioral changes produced by acute stress on locomotor activity and behavioral despair in rats at 12-weeks post-ovariectomy. Ovariectomized female Wistar rats were assigned to eight groups: vehicle group (10% DMSO), three groups with chrysin and three groups with the same dose of allopregnanolone (0.5, 1, and 2 mg/kg), and one group with diazepam (2 mg/kg). The treatments were administered for seven consecutive days and the effects were evaluated in the locomotor activity and swimming tests. Chrysin (2 mg/kg) increased the latency to first immobility and decreased the total immobility time in the swimming test as the reference drugs allopregnanolone and diazepam (2 mg/kg); while locomotor activity prevented the behavioral changes produced by swimming. In conclusion, chrysin exerts a protective effect against the behavioral changes induced by acute stress, similarly to the neurosteroid allopregnanolone and the benzodiazepine diazepam in rats subjected to a surgical menopause model.

## 1. Introduction

Stress is highly experienced in the world’s population. Stressors produced in several environments in which the individual develops, such as family, work, academic and socio-environmental, lead to the development of serious psychiatric disorders, among which post-traumatic stress disorder, anxiety and depression stand out [1,2]. The same effects of stress have been identified at preclinical research [3,4].

Female organisms are highly vulnerable to stress, due in part to the hormonal cyclicality they present throughout their life span [5,6], since hormones produce biological effects in several regions of the central nervous system (CNS) and play an important role in behavior and cognition [7,8]. In this way, a reduction of plasma and brain concentration of steroid hormones such as estradiol, progesterone, and their reduced metabolite allopregnanolone is associated with irritability, anxiety, and depression symptoms, particularly during premenstrual, postpartum and transition to menopause period in women [9,10]. These same effects occur as a consequence of ovarian hormones decline induced by surgical menopause [11].

Menopause marks the end of a woman’s reproductive period, characterized by ovarian dysfunction and the subsequent reduction of plasmatic concentrations of steroid hormones. This physiological state is associated to a greater vulnerability to the negative effects of stress, which increases the incidence and severity of the symptoms of irritability, anxiety, and depression [12,13,14]. Depression is a common psychiatric disorder, involving persistent sadness or loss of pleasure and suicidal behavior that impair daily functioning [15]. Additionally, it was ranked by World Health Organization as the single largest contributor to global disability and the major contributor to suicide deaths [16]. These changes in emotional state are more severe when a woman undergoes surgical remotion of one or both ovaries, knowns as oophorectomy, which triggers a state called “surgical menopause” [17,18]. Interestingly, the effects produced by the low concentration of ovarian hormones as a consequence of surgical remotion of ovaries in women have been reproduced in preclinical studies using long-term ovariectomy in rats [19,20]. This allows exploration of the potential therapeutic effects of several molecules on neuropsychiatric symptoms associated with surgical menopause in women [21].

Serotonin reuptake inhibitors such as fluoxetine and, in some cases, benzodiazepines such as diazepam are used to treat some symptoms of menopause [22,23]. However, patients can recur to therapeutic alternatives based on active molecules of vegetal origin such as flavonoids [24,25]. Regrettably, there is a lack of controlled studies to support or refute their potential therapeutic actions [26]. In this sense, some natural molecules, such as flavonoids, produce anxiolytic-like effects in animal models, comparable to those produced by diazepam [27,28]. At preclinical research, the flavonoid chrysin has verified anxiolytic-like effects mediated through the GABA_A_/Benzodiazepine receptor complex [29,30]. However, unlike benzodiazepines, chrysin exert anxiolytic actions without inducing sedation or muscle relaxation [30,31]. The anxiolytic effects of chrysin appear after a single injection of this flavonoid in rats with 12-weeks post-ovariectomy [32]. Both acute (single injection) and chronic (21 consecutive days) effects of chrysin (2, 4, 8 mmol/kg) on anxiety-like behavior are related to Fos immunoreactivity in the lateral septal nucleus [33], a brain structure involved in the physiopathology of anxiety and depression disorders. In addition, chrysin (1, 5, and 10 mg/kg for 28 days) produces antidepressant-like effects in the swimming test in male Wistar rats [34], and these effects are similar to those produced by steroid hormones progesterone and allopregnanolone, in post-menopausal rats [35]. However, it is unknown if the pre-treatment during seven days with chrysin can block the establishment of behavioral despair triggered by 15 min-swimming of post-ovariectomized rats as it occurs with progesterone and allopregnanolone [35,36,37,38]. Therefore, the present study explored the potential protective effect of the flavonoid chrysin against the acute stress induced by the swimming test in rats with chronic absence of ovarian hormones and it was compared with the effects produced by allopregnanolone and diazepam, two substances with proven anxiolytic and anti-stress effects at both preclinical and clinical research.

## 2. Results

### 2.1. Locomotor Activity Test

Analysis of crossing showed statistical differences according to session factor [F(1,64) = 29.885; *p* < 0.001], this variable was significantly decreased in test compared to the pre-test session. There were no differences between treatments [F(7,64) = 1.178; *p* = 0.328, NS], but the interaction between factors was significant [F(7,64) = 2.346; *p* < 0.034]. Rats administered with vehicle, allopregnanolone and chrysin rats’ groups (both 0.5 mg/kg) displayed less crossing in test respect to pre-test, while the remaining groups did not have any change in crossing between sessions. During the test session, chrysin (0.5 mg/kg), showed low values compared to the vehicle group and produced less crossing compared to diazepam, allopregnanolone, and chrysin (2 mg/kg). See Table 1A.

The time spend in grooming did not vary between sessions [F(1,64) = 1.029; *p* = 0.314, NS] neither due to treatments [F(7,64) = 1.476; *p* = 0.192, NS]. Similarly, interaction between factors was not statistically significant [F(7,64) = 2.250; *p* = 0.041] (see Table 1B).

Rearing had differences due to session [F(1,64) = 43.082; *p* < 0.001] where rearing was lower in the test compared to the pre-test session. Pharmacological treatments had no effects in rearing [F (7,64) = 0.768; *p* = 0.616, NS]. On the other hand, the interaction between factors was significant [F(7,64) = 3.366; *p* < 0.004]. The groups vehicle, allopregnanolone (0.5, 1, 2 mg/kg) and chrysin (0.5) had less rearing in test compared to the respective pre-test session, while the groups chrysin (1, 2 mg/kg) and diazepam had no differences in rearing between sessions. In the test session, the dose of 0.5 mg/kg of chrysin had less rearing compared to the doses of 2 mg/kg of the same drug. No changes were observed in the other groups during test (Table 1C).

### 2.2. Forced Swim Test

#### 2.2.1. Latency to the First Immobility (s)

A statistical analysis of latency to the first period of immobility showed significant differences between treatments [F(7,64) = 19.237; *p* < 0.001], as latency was higher in diazepam, chrysin and allopregnanolone groups (2 mg/kg) compared to the vehicle. The factor session was also statistically significant [F(1,64) = 40.965; *p* < 0.001], and latency decreased from the pre-test to test session. The interaction between factors was not significant [F(7,64) = 1.445; *p* = 0.203, NS]. See Table 2.

#### 2.2.2. Total Time of Immobility (s)

Immobility time was modified by treatments [F(7,64) = 6.269; *p* ≤ 0.001], but not by session [F(1,64) = 3.125; *p* = 0.082, NS]. The interaction between factors was also significant [F(7,64) = 3.589; *p* = 0.003]. A post hoc test revealed that the total time of immobility increased from pre-test to test in vehicle and allopregnanolone treated groups (0.5 mg/kg), while it decreased in the diazepam group. The effect of treatments was observed in test session where chrysin (1 and 2 mg/kg), neurosteroid allopregnanolone (2 mg/kg) and benzodiazepine diazepam had shorter times of immobility, with respect to the vehicle-treated group (Figure 1).

## 3. Discussion

The present study explored the possible protective effects of seven days of pretreatment with flavonoid chrysin against the stress induced by forced swim in rats with long-term absence of ovarian hormones. Effects of chrysin were compared to those produced by allopregnanolone and diazepam. The 15 min-forced swim pre-test represented a stressor that reduced locomotion, exploration, and motivation 24 h later during the 5 min test session, these effects were prevented by the treatment with chrysin, similarly to allopregnanolone and diazepam. The results suggest that chrysin exerts anti-stress effects that block the development of behavioral despair triggered by forced swimming in rats with long-term absence of ovarian hormones.

In preclinical research, ovariectomy in rodents was used as a surgical postmenopausal model that produces changes equivalent to oophorectomy in women [39]. Ovariectomized rats develop endocrine, physiological, and emotional alterations characterized by an early increase in follicle-stimulating hormone levels and a decrease in progesterone and estrogen levels [40], similar to women who undergo oophorectomy [41,42,43]. The long-term effects of ovariectomy are accompanied by the loss of bone density [44], and reduced concentrations of dopamine, serotonin, testosterone, estradiol, allopregnanolone and GABA, and increased concentrations of noradrenaline and cortisol in the brain [43,45]. All those changes produce neuroanatomical modifications in brain structures involved in stress, anxiety, and depression such as the raphe nucleus, hippocampus, prefrontal cortex, hypothalamus, amygdala, and lateral septal nucleus [10,43]. Additionally, an increase in anxiety- and depression-like behavior is detected several weeks post-ovariectomy [20,39], which resemble the phenomena observed in women [11]. These findings validate the utility of the rat ovariectomy as a menopause model for exploring the effects of molecules like the flavonoid chrysin on behavior, which could be an alternative treatment due to its accessibility and low sedative activity.

A contribution of the present study is the exploration of the effects of chrysin in ovariectomized rats 12-weeks after the surgery, when ovarian hormones are decreased [39,46] and the organism is more vulnerable to the deleterious effects of stress [12,47]. Even under this vulnerable condition chrysin exerted protective effects against the stress induced by forced swim pre-test in a similar way than allopregnanolone and diazepam. It suggests a potential therapeutic effect of this flavonoid against the stress related disturbances on the CNS without the secondary effects of benzodiazepines which still in anxiolytic doses increased risk of developing dementia [48,49], road accidents [50,51], desensitization of GABA_A_ receptors in chronic treatments [52], and development of pharmacological tolerance which leads to the loss of their therapeutic capacity [53]. Interestingly, in previous studies it has been reported that flavonoid chrysin even in higher doses (>10 mg/kg) devoid of the characteristics side effects of benzodiazepines as diazepam, including sedation, motor incoordination and cognitive impairment [30,31]. This is the principal reason to consider that chrysin could be a potential alternative to benzodiazepines in the control of anxiety symptoms, particularly in individuals with low concentrations of ovarian hormones as occur in patients cursing natural and surgical menopause.

Locomotor activity explores the motor effects of several treatments as secondary metabolites and plant extracts [54,55,56] or effects associated to physiological states such as estrous cycles [57,58], menopause [20], pregnancy and breastfeeding [59]; or even the interaction of both treatments and physiological states [32,35]. Crossing is considered an indicator of activity, which is influenced by several factors and treatments. Stressors as chronic unpredictable stress (CUS) and space restriction decreases crossing compared to unstressed animals [60,61], which resembles our findings where animals treated with vehicle decreased crossing after the pre-test of forced swim while the treatments with chrysin, allopregnanolone or diazepam blocked this decrease, suggesting a protective effect against stress. Filho and collaborators have previously observed a protector effect of the administration of chrysin (28 days) against the stress induced by CUS on motor activity, which is associated to neuroplastic changes in female mice [62]. The present study confirms this effect and suggests that chrysin only requires seven days of treatment to exert an anti-stress effect and probably block the establishment of despair behavior in ovariectomized rats.

Grooming is a behavior with a high motivational component, sensible to several stressors [63,64] and anxiolytic and antidepressant drugs [65]. Grooming can decrease with severe stressors, depending on the intensity and duration of the stress triggered [66,67]. Thus, short grooming times are considered related to behavioral despair states [68]. In the present study, vehicle treated rats decreased grooming from pre-test to test, after being exposed to forced swim, which did not occur in animals treated with chrysin, allopregnanolone and diazepam, supporting again the protective effects of these drugs against stress as have been previously observed with other anxiolytic and antidepressant treatments such as 0.09 mg/kg of genistein or 17β-estradiol in ovariectomized rats [32,46].

Rearing is interpreted as an indicator of exploration [69,70]. This exploration is related to socioenvironmental factors that trigger arousal states in response to predators, food, or other information from the environment [71]; however, this behavior is also reduced by ovariectomy and some stressors [20,63], while anxiolytic and antidepressant drugs restore it to control values [32,35]. Similarly, in this present study, swimming-induced stress decreased rearing in vehicle-treated animals in test sessions compared to the pre-test session indicating the negative effect of stress; however, chrysin and the other treatments prevented this change, showing similar values in this variable between pre-test and test session, which indicate the protective effect of treatments against stress induced by the forced swim test. Similar findings on rearing have been reported with stressors as space restriction and predator odors [67] and other stressors such as space restriction in cold water [72]. Interestingly, reduction in rearing behavior is restored by anxiolytic and antidepressant drugs [32,35], as it occurs with flavonoid chrysin in the present study.

Forced swim test measures the total time of immobility as an indicator of a “behavioral despair” useful to predict potential antidepressant-like effects of drugs [73] and the impact of several stressors such as food restriction [74], among others. The latency to the first immobility assesses the magnitude of the first animal effort to cope with the aversive situation of being forced to swim without escape [75]. Total time of immobility is interpreted as an indirect measure of motivation, the longer the immobility, the lesser the motivation [73]. These behaviors are associated to changes at neurochemical and electrophysiological levels caused by stress and relatively equivalent to those observed in human depression [76,77,78] and in structures that modulate motivation as the lateral septum [79,80]. These changes are reverted with the administration of antidepressant drugs [37,81].

The present study showed the establishment of “behavioral despair” as latency decreased and total time of immobility increased during the test session [37,73,82,83] in the vehicle and chrysin 0.5 mg/kg groups, indicating that this dose of chrysin is devoid of an antidepressant-like effect. Considering that treatments in the present study were injected during seven consecutive days before the pre-test session in the forced swim test, if treatments exerted an anxiolytic, antidepressant or antistress-like effect, it would be expected that the total time of immobility does not increase in the test session, as it occurred in the present study with chrysin and allopregnanolone 1 and 2 mg/kg, and diazepam 2 mg/kg. These findings again support the protective effects of treatments against stress induced during the pre-test session in the forced swim test. Contrarily, vehicle and chrysin 0.5 mg/kg groups increased the total time of immobility because they lack a protective treatment against stress induced by the 15 min pre-test, therefore these groups develop despair behavior (increase of immobility).

Finally, one limitation of the present study was that we did not explore the mechanism of action of the behavioral effects produced by flavonoid chrysin in rats with long-term absence of ovarian hormones. However, it is widely known that chrysin exerts its principally pharmacological action mainly through the GABA_A_ receptors. In Xenopus laevis oocytes, chrysin binds on α1, β1, and γ2 subunits of the GABA_A_ receptor, which could be associated with its anxiolytic-like actions [84]. In this way, it is probable that flavonoid chrysin acted through GABA_A_ receptor increasing chloride ions influx in neurons, which relates to improvements in stress coping, anxiolytic-like effects, and even antidepressant-like effects in the forced swim test [35], as it occurs with GABAergic compounds, including neurosteroids as progesterone and allopregnanolone exerting protective effects against stress and behavior despair [38,83,85,86]. In support, it has been previously reported that pretreatment with picrotoxin, bicuculline and flumazenil, antagonists of the GABA_A_ receptor, block the anxiolytic- and antidepressant-like effects of chrysin [30,32,35]. In this sense, it is to possible suggest that the GABA_A_ receptor is therefore a molecular target to the substances tested in the present experiment, chrysin [30,31], allopregnanolone [37,87], and diazepam [88], without discarding the participation of other systems of neurotransmission.

Altogether, the present results indicate that flavonoid chrysin prevents the establishment of anxiety and despair-like behaviors induced by a severe acute stressor in rats subjected to the surgical menopause model, like neurosteroid allopregnanolone and benzodiazepine diazepam. The present findings prompt further clinical studies on the neuropsychiatric effects of chrysin to contribute to the prevention and control of anxiety and depression symptoms in women under surgical menopause or transition to natural menopause, with finality of improving life quality.

## 4. Material and Methods

### 4.1. Ethics

All experimental procedures were performed according to the Guide for the Care and Use of Laboratory Animals published by the National Institutes of Health [89] and the Norma Official Mexicana para el Uso y Cuidado de Animales de Laboratorio [90]. All efforts were made to minimize animal discomfort and the number of animals according to the Reduce, Refine, Replace (3R) principles of preclinical research [91]. The general protocol was approved by the Institutional Internal Committee for the care and use of laboratory animals of the Escuela de Medicina Veterinaria y Zootecnia de la Universidad de Tlaxcala (UTx/MVZ-189/12).

### 4.2. Animals

Seventy-two female Wistar rats (three-month-old; 250–300 g at the beginning of the experiments) were housed in Plexiglas cages, 4 rats per cage (44 cm width × 33 cm length × 20 cm height), with a 12 h/12 h light/dark cycle (lights on at 07:00 h), average room temperature of 25 °C (±1 °C) and free access to purified water and food (Purina Pellets, Agribrands Purina Mexico, Mexico City, Mexico) during the study period. All experimental sessions were conducted between 09:00 and 12:00 h.

### 4.3. Drugs

Pentobarbital sodium was purchased from Cheminova de Mexico, Mexico City, Mexico. Reg. SAGARPA Q-7048-044). Benzalkonium chloride from Medipharm^®^, San Luis Río Colorado, Sonora, Mexico. Dipirona50^®^ was obtained from Virbac Animal Health, Guadalajara, Mexico. Dimethylsulfoxide (DMSO) was purchased from Golden Bell Reactivos (Mexico City, Mexico). Atropine sulphate, chrysin (Chry, 5,7-Dihydroxyflavone < 97%) and allopregnanolone (Allo, 5α-Pregnan-3α-ol-20-one) were purchased from Sigma-Aldrich (St. Louis, MO, USA). Diazepam (Valium, injectable solution) was obtained from Laboratory Roche (Mexico City, Mexico). Saline solution (0.9%) was purchased from PiSA Farmacéutica (Guadalajara, Jalisco, Mexico).

### 4.4. Ovarectomy

Only females with three continuous regular cycles (4–5 days), verified daily by vaginal smears [92], were included in the study. Surgical remotion of both ovaries was performed through abdominal ventral incision under deep anesthesia (pentobarbital sodium, 60 mg/kg, i.p.) and previous administration of atropine sulphate (0.05 mg/kg, i.p.). After remotion of both ovaries the area was carefully cleaned with benzalkonium chloride, and muscle and skin were sutured separately. Analgesic and antipyretic medication (Dipirona50^®^, 50 mg/kg, i.m.) was administered after the surgery and during the subsequent four days to minimize post-surgical pain. After surgery and during the entire experimental protocol all the rats were examined daily to detect health anomalies, including changes in water and food intake, eye orbital tightening, nose/cheek flattening, ear position, vibrissae position, hair bristling, and changes in coat color and texture. Some of these evaluations were based on the Rat Grimace Scale [93]. Then, rats were randomly assigned to each of the experimental groups and returned to the housing facilities for 12 weeks to assure the long-term absence of ovarian hormones and behavioral indicators of anxiety- and despair-like states [32,39]. After this time all groups started treatments and were subjected to behavioral tests.

### 4.5. Experimental Groups

Eight experimental groups (n = 9 each group) were assembled: a vehicle group (10% DMSO), three groups with chrysin and three groups with allopregnanolone (0.5, 1 and 2 mg/kg), and one diazepam group (2 mg/kg). After 12 weeks post-ovariectomy, drugs were injected via i.p. for seven consecutive days. One hour after the last administration rats were submitted to locomotor activity tests (5 min) and subsequently to forced swim stress (pre-test session 15 min, stress induced session). Twenty-four hours later all groups were evaluated for 5 min in each of the same behavioral tests. Diazepam (2 mg/kg) was used as a reference anxiolytic drug [32,94], which also reduces total time of immobility in the swimming test [33]. The doses of chrysin were based on previous works in which doses from 1 to 4 mg/kg had anxiolytic-like effects [32,33] and 1 and 2 mg/kg had antidepressant-like effects in male rats [35]. Therefore, since female rats are more responsive to anxiolytic and antidepressant drugs, a specific doses-response curve (0.5, 1 and 2 mg/kg) was included to identify possible anxiolytic- and antidepressant-like effects using lower doses than those used in male rats under our experimental conditions. Allopregnanolone doses were used as controls of antidepressant-like agents with GABAergic properties [35,37].

### 4.6. Behavioral Tests

#### 4.6.1. Locomotor Activity Test (LAT)

The changes in spontaneous locomotor activity were evaluated by individually placing the rats into an opaque Plexiglas cage (44 *×* 33 cm, base; 20 cm high) with the floor delineated into 12 squares (11 × 11 cm). Variables as number of crossing and time in seconds spent in grooming and rearing were evaluated. Rats were considered to have crossed from one square to another (crossing) when the hind legs crossed the line from one square to another. Grooming included all self-directed behaviors of cleaning from head, ears, limbs, and anal–genital region; this behavior may be significantly affected by diverse stressors [46]. Rearing was measured when rats explored the cage in a vertical position standing on its rear limbs. Rearing was evaluated considering that is a behavior related with exploration and it is significantly affected by ovariectomy and stressors [20], while anxiolytic and antidepressant drugs may prevent said effects and maintain or increase this variable respect to control groups [32,35]. After each rat was tested, the locomotor activity cage was carefully cleaned with 10% alcohol solution to remove the scent of the previously evaluated rat. After the locomotor activity test, rats were subjected to the swimming test. Approximately 2 min elapsed between tests.

#### 4.6.2. Forced Swim Test (FST)

After LAT, rats were individually forced to swim in a rectangular pool (50 × 30 × 60 cm) with 25 cm deep water (25 ± 1 °C). The variables evaluated were the latency to first immobility and total time of immobility. Latency to the first immobility is the elapsed time since the rat was introduced to the pool, until the first immobility episode. The immobility was considered when the rat floated for more than 1 s without making vigorous movements leading to displacements and only maintaining its head above the water surface. These behaviors are indicators of despair and antidepressant-like effects of clinically effective antidepressant drugs and other substances [73,75]. All experimental sessions were video recorded and two independent observers, blind to treatments, measured the behavioral variables with a concordance level of at least 95%.

##### Forced Swim Pre-Test (Acute Stressor)

As part of the methodology of the forced swimming model, there is the “pre-test” session, which consists of a 15 min session in which the rat is placed in a pool of water that constitutes an aversive stimulus with no possibility of escape [73]. Being a novel environment for the rat as well as aversive, this “pre-test” has been considered a highly stressful situation, which 24 h later, produces a state of hopelessness characterized by a significant increase in the total time of immobility [95,96]. One of the first studies to test this approach was that of Franco Borsini, in which similar duration of immobility behavior, suggestive of hopelessness, was identified between rats that received the 15 min “pre-test” session in forced swimming and rats that were subjected to 3 stressful situations: space restriction, exposure to low temperatures (4 °C) and continuous electric shocks to the paws [82]. Additionally, the severe acute stress of swimming triggers a marked increase in blood glucocorticoid concentration similar to the increase produced by a session of space restriction [97]. Similarly, glucocorticoid levels increase when rats are submitted to forced swim test for 25 min or until complete exhaustion [98]. Likewise, GABA_A_ receptor expression has also been shown to change with chronic exposure to swimming stress [99]. Therefore, in this study, we performed the behavioral measurement of the effects of the forced swim pre-test session, videotaping and analyzing only the first 5 min of this pre-test [100,101] and compared to the 5 min test 24 h later.

### 4.7. Statistical Analysis

The statistical analysis was conducted using Sigma Plot 12 software. Data were analyzed using two-way ANOVA for repeated samples, with treatments and tests session as factors. Data were transformed to ranks using Sigma Plot 12 software to satisfy the assumptions of the normality test and equal variance test, and then analyzed. Values of *p* ≤ 0.05 in the ANOVA were followed by the Student–Newman–Keuls post hoc test. The results are expressed as mean ± standard error.

## 5. Conclusions

The flavonoid chrysin blocks the establishment of anxiety- and despair-like behaviors induced by the forced swim pre-test in ovariectomized rats, resembling the effect of GABAergic drugs as allopregnanolone and diazepam. These results support the potential therapeutic effects of flavonoid chrysin in stress-related disorders like anxiety and depression associated with surgical menopause in females.

## Figures and Tables

**Figure 1 molecules-28-00587-f001:**
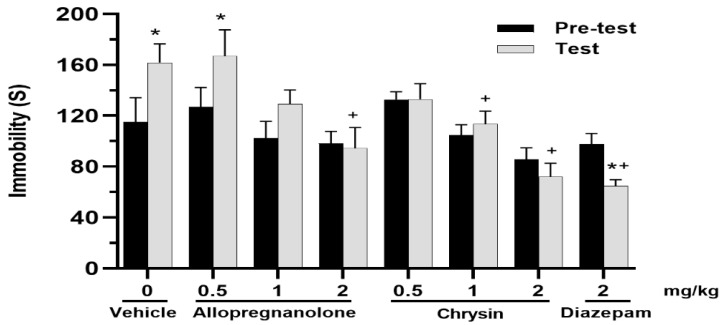
Total time of immobility. Immobility increased from pre-test to test session in vehicle and allopregnanolone (0.5 mg/kg) groups, while no changes were detected in allopregnanolone (1 and 2 mg/kg) and chrysin (all evaluated doses) groups. In the test session, chrysin (1 and 2 mg/kg), allopregnanolone (2 mg/kg) and diazepam significantly reduced the immobility time with respect to vehicle in the test session. * *p* < 0.05 vs. same group in pre-test session, + *p* < 0.05 vs. vehicle in the test session. Data are represented as mean ± standard error. Two-way repeated measures ANOVA, post hoc Student–Neuman–Keuls.

**Table 1 molecules-28-00587-t001:** Variables evaluated in the locomotor activity test.

Variable/Group (mg/kg)	Pre-Test	Test	Totals
(A) Crossing (s)			
Vehicle	54.39 ± 9.02	31.94 ± 6.00 ^+^	43.17 ± 5.92
Chrysin			
0.5	51.00 ± 2.39	23.22 ± 3.60 ^+^*	37.11 ± 3.97
1.0	50.00 ± 3.05	43.11 ± 3.07	46.56 ± 2.26
2.0	54.56 ± 3.10	45.44 ± 3.80	50.00 ± 2.62
Allopregnanolone			
0.5	53.61 ± 6.35	32.61 ± 4.27 ^+^	43.11 ± 4.50
1.0	46.44 ± 4.16	43.23 ± 7.04	44.84 ± 3.99
2.0	57.61 ± 8.70	50.00 ± 8.16	53.81 ± 5.86
Diazepam			
2.0	51.67 ± 2.98	48.56 ± 4.01	50.11 ± 2.45
Totals	52.41 ± 1.91	39.77 ± 2.06 ^#^	
(B) Grooming (s)			
Vehicle	30.16 ± 5.95	11.67 ± 3.52	20.92 ± 4.03
Chrysin			
0.5	19.30 ± 2.55	10.68 ± 1.52	14.99 ± 1.78
1.0	20.40 ± 1.97	18.49 ± 1.54	19.44 ± 1.24
2.0	23.43 ± 3.24	20.49 ± 1.71	21.96 ± 1.81
Allopregnanolone			
0.5	15.35 ± 5.63	21.70 ± 10.52	18.53 ± 5.84
1.0	14.89 ± 5.50	26.20 ± 7.54	20.54 ± 4.73
2.0	23.68 ± 9.55	32.96 ± 11.59	28.32 ± 7.37
Diazepam			
2.0	23.12 ± 2.58	27.31 ± 3.05	25.21 ± 2.00
Totals	21.29 ± 1.83	21.19 ± 2.32	
(C) Rearing (s)			
Vehicle	42.28 ± 5.52	24.65 ± 3.62 ^+^	33.46 ± 3.85
Chrysin			
0.5	39.28 ± 2.32	21.14 ± 1.65 ^+^*	30.21 ± 2.60
1	37.65 ± 1.76	36.28 ± 2.53	36.96 ± 1.51
2	37.35 ± 1.59	38.26 ± 3.55	37.80 ± 1.89
Allopregnanolone			
0.5	48.90 ± 6.21	23.21 ± 3.97 ^+^	36.06 ± 4.74
1.0	45.50 ± 4.92	28.85 ± 4.34 ^+^	37.17 ± 3.77
2.0	46.62 ± 6.77	40.35 ± 12.18 ^+^	43.49 ± 6.80
Diazepam			
2.0	35.95 ± 2.46	33.25 ± 3.00	34.60 ± 1.91
Totals	41.69 ± 1.58	30.75 ± 1.96 ^#^	

(A) ^#^
*p* < 0.001 vs. pre-test; ^+^
*p* < 0.05 vs. same group pre-test; * *p* < 0.05 vs. Allopregnanolone 2 mg/kg, Chrysin 2 mg/kg, and Diazepam within same session. (B) No significant differences in grooming were found between sessions and groups. (C) ^#^
*p* < 0.01 vs. pre-test; ^+^
*p* < 0.05 vs. same group pre-test; * *p* < 0.05 vs. Chrysin 2 mg/kg within same session. Data are represented as mean ± standard error. Two-way repeated measures ANOVA, post hoc Student–Neuman–Keuls.

**Table 2 molecules-28-00587-t002:** Latency to the first immobility (s).

Groups/mg/kg	Pre-Test	Test	Totals
Vehicle	58.92 ± 6.19	55.11 ± 16.86	57.02 ± 8.72
Chrysin			
0.5	53.34 ± 4.04	30.93 ± 3.18	42.14 ± 3.69
1	73.03 ± 4.62	21.88 ± 3.80	47.45 ± 6.85
2	130.12 ± 10.05	81.57 ± 18.25	105.85 ± 11.70 *
Allopregnanolone			
0.5	56.36 ± 6.54	36.40 ± 7.88	46.38 ± 5.52
1	96.31 ± 7.46	55.93 ± 5.01	76.12 ± 6.55
2	128.35 ± 11.52	91.38 ± 19.44	109.86 ± 11.84 *
Diazepam			
2	154.01 ± 12.71	93.50 ± 14.13	123.76 ± 11.78 *
Totals	93.81 ± 5.21	58.34 ± 5.29 ^+^	

* *p* < 0.001 vs. Vehicle; ^+^
*p* < 0.001 vs. pre-test. Data are presented as mean ± standard error. Two-way ANOVA for repeated measures, post hoc Student–Newman–Keuls.

## Data Availability

The raw data from this study are available from the corresponding author upon reasonable request.

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
