# Peer review of "Behavioral Despair Is Blocked by the Flavonoid Chrysin (5,7-Dihydroxyflavone) in a Rat Model of Surgical Menopause"

_molecules, 2023, doi:10.3390/molecules28020587_

Round 1

Reviewer 1 Report

-       Minor revisions to be revised (see the comments in the pdf file)

-       Why are the authors interested in the rearing of the rats? Do they suspect that the tested compounds have a side effect on the rats by enhancing their rearing? If so, I think that the authors should talk about this either in the introduction or the discussion.

-       Why the immobility time increased in the test compared to the pre-test in the vehicle group and in the 0.5 mg/kg allopregnanolone group? And this behavior is not seen in the other groups?

-       In all the experiments, it is clearly seen that the concentration of 0.5mg/kg has no effect whatsoever. Why the authors attributed this concentration in their study? Is it a common dosage used in treatments? And if so, why do they think that in their experiments this concentration showed no effect?

-       The tested flavonoid (chrysin) has an effect comparable to allopregnanolone and diazepam, yet diazepam has the better effect whether it is related to the immobility time, the latency to immobility or even in the locomotor activity variables. Thus, can the author explain the reason why it is better to use chrysin? 

-       In the discussion section, the authors clearly state the secondary effects of benzodiazepines on the CNS, and they state that flavonoids don’t have the same secondary effects. This said, in this article, they never showed if the chrysin really do not have the same secondary effects as benzodiazepine. So, I suggest the authors either to cite papers that shows this phenomenon or if they can do some molecular experiments showing the absence if these secondary effects.

-       The authors want to show that chrysin has an anti-stress, anti-depression effect. This said, the authors used several behavior test to show this effect but in my opinion there are other tests that could have been done to emphasize the anti-stress and anti-depression effect like the sucrose preference test, the tail suspension test, the light/dark box and the stress induced hyperthermia, etc.. So, can the authors explain why did they choose only the tests that they mentioned in the article and no other?

-       In the discussion section (line 344), the authors speculate that the GABAA receptors are the targeted receptors of the chrysin, yet there is no experiment in the article that shows this matter. Either the authors should explain, in detail, how did they conclude this aspect by citing articles that proves this process or do some molecular testing (like FRET experiments that will show the interaction between the chrysin and the GABAA receptor).

-          Line 154: “one instead of on” square

-          Line 199 to 200: The phrase is not clear and it need to be considered

-          Line 205: Chrysin rats’ group instead of chrysin

-          Line 205: The term “similar” is misused. I think the author meant “compared to the vehicle group”

-          Line 218: In respect to instead of in respect

-          Line 316: in this present and not in present. Also add a coma after the word study

-          Line 317: In respect instead of respect

-          Line 344: Use therefore instead of then

Author Response

Dear reviewer,

We thank you for the time spent in reviewing our proposal and the thoughtful comments on our manuscript, which have greatly improved the quality and understanding of our paper. We have revised the manuscript according to the reviewers suggestions. Please find our corresponding response to your comments below.

-   Minor revisions to be revised (see the comments in the pdf file)

Response. Thank you for the corrections; they have been included in the revised manuscript.

- Why are the authors interested in the rearing of the rats? Do they suspect that the tested compounds have a side effect on the rats by enhancing their rearing? If so, I think that the authors should talk about this either in the introduction or the discussion.

Response. Rearing behavior is considered as indicator of exploration, which may be affected by ovariectomy and stress as the forced swim test. In this way, in present study rearing was evaluated to identify potential changes produced by stress induced by the forced swim test and potential protective effect of chrysin, allopregnanolone, and diazepam. It has been identified that ovariectomy (Puga-Olguin et al., 2019) and some stressors (Kalueff and Touhimaa, 204) reduces rearing, while some anxiolytic and antidepressant drugs maintain or increase this behavior in stressed animals (Rodríguez-Landa et al., 2020; Cueto-Escobedo et al., 2020). Therefore, it was included as an additional parameter to identify the protective effects of treatments against stress induced by the forced swim test in the ovariectomized rats. This information has been included in methods and discussion, respectively, as reviewer indicated, see page 157-163, lines 327-334, 336-338.

Puga-Olguín, A.; Rodríguez-Landa, J.F.; Rovirosa-Hernández, M.J.; Germán-Ponciano, L.J.; Caba, M.; Meza, E.; Guillén-Ruiz, G.; Olmos-Vázquez, O.J. Long-term ovariectomy increases anxiety- and despair-like behaviors associated with lower Fos immunoreactivity in the lateral septal nucleus in rats. Behav. Brain Res. 2019, 360, 185–195. doi:10.1016/j.bbr.2018.12.017.

Rodríguez-Landa, J.F.; Hernández-López, F.; Cueto-Escobedo, J.; Herrera-Huerta, E.V.; Rivadeneyra-Domínguez, E.; Bernal-Morales, B.; Romero-Avendaño, E. Chrysin (5,7-dihydroxyflavone) exerts anxiolytic-like effects through GABAA receptors in a surgical menopause model in rats. Biomed. Pharmacother. 2019, 109, 2387–2395. doi:10.1016/j.biopha.2018.11.111.

Cueto-Escobedo, J.; Andrade-Soto, J.; Lima-Maximino, M.; Maximino, C.; Hernández-López, F.; Rodríguez-Landa, J.F. Involvement of GABAergic system in the antidepressant-like effects of chrysin (5,7-dihydroxyflavone) in ovariectomized rats in the forced swim test: Comparison with neurosteroids. Behav. Brain Res. 2020, 386, 112590. doi:10.1016/j.bbr.2020.112590.

Kalueff, A.V.; Tuohimaa, P. Grooming analysis algorithm for neurobehavioural stress research. Brain Res. Protoc. 2004, 13, 151–158. doi:10.1016/j.brainresprot.2004.04.002.

-       Why the immobility time increased in the test compared to the pre-test in the vehicle group and in the 0.5 mg/kg allopregnanolone group? And this behavior is not seen in the other groups?

Response. As we mentioned in 2.6.2.1 section, in this model of behavioral despair, pretest session of 15 min is a stressful situation which induces “despair” in the rats (increasing total time of immobility) when animals are evaluated again 24 h later with a 5 min-test. Considering that treatments in present study were injected during seven days before behavioral test, if treatments exert anxiolytic, antidepressant and antistress-like effects it is to be expected that the total time of immobility remain unchanged in the test session, as it occurs in present study with chrysin and allopregnanolone 1 and 2 mg/kg, and 2 mg diazepam. Contrarily, immobility increased in vehicle group because it was not protected by   treatments against the stress induced by 15 min pretest, therefore this group developed behavioral despair. This same effect occurred with 0.5 mg/kg of chrysin, which indicate that this dose lacks protective effects. This information has been included in discussion, see lines 349-360.

-       In all the experiments, it is clearly seen that the concentration of 0.5 mg/kg has no effect whatsoever. Why the authors attributed this concentration in their study? Is it a common dosage used in treatments? And if so, why do they think that in their experiments this concentration showed no effect?

Response. In previous studies chrysin 0.5 mg/kg produces anxiolytic-like effects in the elevated plus maze, however, this effect is reported in male rats. Considering that the present study included ovariectomized female rats we used a dose-response curve to identify if the same effects are detected in male and female rats at the same doses. In this case, the no effect of chrysin 0.5 mg/kg in present study could be related with sex and physiological condition of the subjects (a reduced concentration of ovarian hormones produced by ovariectomy). For this reason, we considered important to evaluate the three doses of chrysin in present study. This information has been included in methods, lines 143-148.

-       The tested flavonoid (chrysin) has an effect comparable to allopregnanolone and diazepam, yet diazepam has the better effect whether it is related to the immobility time, the latency to immobility or even in the locomotor activity variables. Thus, can the author explain the reason why it is better to use chrysin?

Response. In previous studies it has been reported that diazepam and other benzodiazepines may produces cognitive impairment, sedation, and motor incoordination even in anxiolytic doses. However, previous studies have discarded these side effects when different doses of chrysin have been evaluated in animal models. Also, benzodiazepines must not be used for long periods.  This is the principal reason to consider that chrysin could be a potential alternative to benzodiazepines in the control of anxiety symptoms. This information has been included in discussion section, lines 295-301.

-       In the discussion section, the authors clearly state the secondary effects of benzodiazepines on the CNS, and they state that flavonoids don’t have the same secondary effects. This said, in this article, they never showed if the chrysin really do not have the same secondary effects as benzodiazepine. So, I suggest the authors either to cite papers that shows this phenomenon or if they can do some molecular experiments showing the absence if these secondary effects.

Response. Thank you for your recommendation, in this reviewed version this issue has been addressed as reviewer indicated. This information has been supported by previous studies that identify not negative effects of chrysin on motor coordination and cognitive function even with higher doses (10 mg/kg) that those used in present study. See, lines 295-301.

-       The authors want to show that chrysin has an anti-stress, anti-depression effect. This said, the authors used several behavior test to show this effect but in my opinion there are other tests that could have been done to emphasize the anti-stress and anti-depression effect like the sucrose preference test, the tail suspension test, the light/dark box and the stress induced hyperthermia, etc. So, can the authors explain why did they choose only the tests that they mentioned in the article and no other?

Response. Thank for your comment. In this study we have included a behavioral test battery that has been standardized and used in our laboratory, which permit identify the anti-stress and antidepressant-like effects of diverse substances. Additionally, the forced swim test is one of the most used animal models for predicting the potential antidepressant-like activity of diverse molecules. In this way, the use of these test allows us compare results with previous reports to support the pharmacological effects of chrysin. However, we cannot discard the use of the test suggested by reviewer in future studies to complement the results here presented.

-       In the discussion section (line 344), the authors speculate that the GABAA receptors are the targeted receptors of the chrysin, yet there is no experiment in the article that shows this matter. Either the authors should explain, in detail, how did they conclude this aspect by citing articles that proves this process or do some molecular testing (like FRET experiments that will show the interaction between the chrysin and the GABAA receptor).

Response. Thank you for your suggestion. This issue has been addressed in this new version and precise information to support this proposal was included. See lines 361-374.

-          Line 154: “one instead of on” square

-          Line 199 to 200: The phrase is not clear and it need to be considered

-          Line 205: Chrysin rats’ group instead of chrysin

-          Line 205: The term “similar” is misused. I think the author meant “compared to the vehicle group”

-          Line 218: In respect to instead of in respect

-          Line 316: in this present and not in present. Also add a coma after the word study

-          Line 317: In respect instead of respect

-          Line 344: Use therefore instead of then

Response. Thank you for your corrections and suggestion, all of them has been corrected and addressed in this version.

Reviewer 2 Report

General comments for Manuscript ID: molecules-2068606

The study evaluated the antidepressant effect of chrysin in rat model of surgical, menopause. The study is an interesting study with good design. I am particularly worried over the nature of conclusion the authors reached in the study: “Seven days of treatment with the flavonoid chrysin are enough to block the deleterious effects on motivation of the stress……” this must be reversed. One week should not appear anywhere in the conclusion. Also, there are no study showing that the mechanisms of action of chrysin-induced antidepressant effect in surgical menopause-induced depression in rats. There was no receptor interaction study or any measurement of GABA concentration, GAD enzyme or GABAA receptor expression to estimate GABAergic neurotransmission. I am wondering why authors failed to perform these assays. I would recommend these assays to be performed if the conclusion is to be adopted. I would also recommend the addition of these references https://doi.org/10.1016/j.brainres.2020.146917  and DOI 10.1007/s12031-020-01664-y for the first paragraph on stress, linking the role of psychosocial stress as a potential source of anxiety and depression or the inferiority that could emanate from menopausal phase and experience of life.

Author Response

Dear reviewer,

We thank you for the time spent in reviewing our proposal and the thoughtful comments on our manuscript, which have greatly improved the quality and understanding of our paper. We have revised the manuscript according to the reviewers suggestions. Please find our corresponding response to your comments below.

- The study evaluated the antidepressant effect of chrysin in rat model of surgical, menopause. The study is an interesting study with good design. I am particularly worried over the nature of conclusion the authors reached in the study: “Seven days of treatment with the flavonoid chrysin are enough to block the deleterious effects on motivation of the stress……” this must be reversed. One week should not appear anywhere in the conclusion. Also, there are no study showing that the mechanisms of action of chrysin-induced antidepressant effect in surgical menopause-induced depression in rats. There was no receptor interaction study or any measurement of GABA concentration, GAD enzyme or GABAA receptor expression to estimate GABAergic neurotransmission. I am wondering why authors failed to perform these assays. I would recommend these assays to be performed if the conclusion is to be adopted. I would also recommend the addition of these references https://doi.org/10.1016/j.brainres.2020.146917 and DOI 10.1007/s12031-020-01664-y for the first paragraph on stress, linking the role of psychosocial stress as a potential source of anxiety and depression or the inferiority that could emanate from menopausal phase and experience of life.

Responses: 

Considering the comment of the reviewer the conclusion has been rewritten to avoid include “seven days” now conclusion is focused on the pharmacological effect of chrysin and potential therapeutic effects. See lines 385-390.

In respect to the mechanism of action involved in the effects of chrsysin we have include information that support the potential participation of GABAergic system in the anxiolytic- and antidepressant-like effect of chrysin in ovariectomized rats. See lines 361-374.

Finally, the suggestion of the reviewer respect to measurement of GABA concentration, GAD enzymes activity or expression of GABAA receptors remains to be explored in future studies, first because in this study the principal objective was identify if chrysin produced anxiolytic- and antidepressant-like effects in the surgical menopause model, and then explore mechanism involved in future studies; and second, because at this moment we have not the condition to perform these measurements. Thank you for your suggestion.

Round 2

Reviewer 2 Report

Although the status of the manuscript is increased, few comments are left unanswered. I would advice they respond to these leftover comments prior to acceptance.

"There was no receptor interaction study or any measurement of GABA concentration, GAD enzyme or GABAA receptor expression to estimate GABAergic neurotransmission. I am wondering why authors failed to perform these assays. I would recommend these assays to be performed if the conclusion is to be adopted. I would also recommend the addition of these references https://doi.org/10.1016/j.brainres.2020.146917  and DOI 10.1007/s12031-020-01664-y for the first paragraph on stress, linking the role of psychosocial stress as a potential source of anxiety and depression or the inferiority that could emanate from menopausal phase and experience of life"

Author Response

Dear reviewer,

We thank you for the time spent in reviewing the response to your comments. We have revised again the manuscript according to the reviewer suggestions. Please find our corresponding response to your comments below.

Although the status of the manuscript is increased, few comments are left unanswered. I would advice they respond to these leftover comments prior to acceptance.

Response. Thank you for your comment. Please find our corresponding response to your comments below.

"There was no receptor interaction study or any measurement of GABA concentration, GAD enzyme or GABAA receptor expression to estimate GABAergic neurotransmission. I am wondering why authors failed to perform these assays. I would recommend these assays to be performed if the conclusion is to be adopted.

Response. As we answered in the first round of revision, the suggestion of the reviewer respect to measurement of GABA concentration, GAD enzymes activity or expression of GABAA receptors remains to be explored in future studies, first because in this study the principal objective was identify if chrysin produced anxiolytic- and antidepressant-like effects in the surgical menopause model, and then explore mechanism involved in future studies; and second, because at this moment we have not the condition to perform these measurements. However, to support the involvement of GABAergic system in the anxiolytic- and antidepressant-like action of chrysin we included information based in previous studies. Please see lines 362-375 and 387-389.

I would also recommend the addition of these references https://doi.org/10.1016/j.brainres.2020.146917 and DOI 10.1007/s12031-020-01664-y for the first paragraph on stress, linking the role of psychosocial stress as a potential source of anxiety and depression or the inferiority that could emanate from menopausal phase and experience of life"

Response. Thank you for your suggestion. Both references have been included as the reviewer indicated. See lines 40 and 41.